# Germination of White and Red Quinoa Seeds: Improvement of Nutritional and Functional Quality of Flours

**DOI:** 10.3390/foods11203272

**Published:** 2022-10-20

**Authors:** Luciano Martín Guardianelli, María Victoria Salinas, Carla Brites, María Cecilia Puppo

**Affiliations:** 1Centro de Investigación y Desarrollo en Criotecnología de Alimentos-CIDCA, Facultad de Ciencias Exactas-UNLP-CONICET, La Plata 1900, Argentina; 2Instituto Nacional de Investigação Agrária e Veterinária, I.P. Av. da República, Quinta do Marquês, 2780-157 Oeiras, Portugal

**Keywords:** quinoa, composition, antioxidants, amino acids, fatty acids, protein structure, starch gelatinization

## Abstract

Quinoa is an Andean grain, classified as pseudocereal and the exploitation of its nutritional profile is of great interest for the cereal-based industry. The germination of quinoa seeds (white and red royal) was tested at 20 °C for different times (0, 18, 24 and 48 h) to select the best conditions for improving the nutritional quality of their flours. Changes in proximal composition, total phenolic compounds, antioxidant activity, mineral content, unsaturated fatty acids and essential amino acids profiles of germinated quinoa seeds were determined. In addition, changes in structure and thermal properties of the starch and proteins as consequence of germination process were analyzed. In white quinoa, germination produced an increase in the content of lipids and total dietary fiber, at 48 h, the levels of linoleic and α-linolenic acids and antioxidant activity increase, while in red quinoa, the component that was mostly increased was total dietary fiber and, at 24 h, increased the levels of oleic and α-linolenic acids, essential amino acids (Lys, His and Met) and phenolic compounds; in addition, a decrease in the amount of sodium was detected. On the basis of the best nutritional composition, 48 h and 24 h of germination were selected for white and red quinoa seeds, respectively. Two protein bands were mostly observed at 66 kDa and 58 kDa, being in higher proportion in the sprouts. Changes in macrocomponents conformation and thermal properties were observed after germination. Germination was more positive in nutritional improvement of white quinoa, while the macromolecules (proteins and starch) of red quinoa presented greater structural changes. Therefore, germination of both quinoa seeds (48 h-white quinoa and 24 h-red quinoa) improves the nutritional value of flours producing the structural changes of proteins and starch necessary for obtaining high quality breads.

## 1. Introduction

Quinoa (Chenopodium quinoa) is an annual plant of the Amaranthaceae family, dicotyledonous, usually herbaceous, which reaches a height of 0.2 to 3.0 m. It has been cultivated as a staple food for many years in the Andean region [1], gaining popularity these days also in the western region. This plant has oval-shaped seeds, with a diameter that ranges between 1.5 and 2.5 mm, which can adopt colors that vary from yellow to black. Usually, this plant is called pseudocereal; however, due to their origin, it is classified as an Andean crop. The main virtue of quinoa lies in its nutritional profile; the protein content in the grain varies between 12–23% on a dry basis and the quantity of lipid can reach up to 8%, which is double of the value of corn [2]. In addition, it contains high amount of calcium, iron, zinc and magnesium, compared to the cereals. These attributes make it to be considered a superfood, also being used by National Aeronautics and Space Administration (NASA) within the controlled ecological life support system [3], with which it had a new resurgence. In addition, due to the nutritional importance of these seeds, the Food and Agriculture Organization of the United Nations (FAO) stated 2013 as the International Year of Quinoa because this crop plays an important role in the eradication of hunger, malnutrition and poverty [4].

In recent times, the process of monitored germination has been used to generate nutritionally improved grains and seeds. Germination occurs in four fundamental stages: (i) Imbibition of water, (ii) Synthesis and activation of enzyme systems, (iii) Degradation of reserve substances and (iv) Elongation of embryo cells and emergence of the radicle [5]. Through germination process, substantial changes are produced in the biochemical composition of the grains: starch reserves are removed by the action of amylase, which acts on the surface of the granule and forms pores; the nitrogen-containing fractions are shifted toward oligopeptides and free amino acids, and the amino acid composition also changes. Triglycerides begin to hydrolyze and the ratio of saturated/unsaturated fatty acids changes; the amount of antinutritional factors (e.g., phytates, trypsin inhibitor, tannins) decreases significantly and bioactive compounds such as phenols, phytosterols, folates, and GABA (γ-aminobutyric acid) increase. Therefore, in sprouted grains, almost all nutrients are fully available and various antioxidants are found in higher concentrations, thus providing the basis for defining sprouts as “functional foods” [6]. There are several works where the effect of germination on various grains and seeds was studied [7,8,9], as well as on quinoa in particular [10,11,12,13]; however, there are few comprehensive works that analyze how germination affects flours physicochemical properties at nutritional and structural levels. Therefore, the purpose of this work was to improve nutritional characteristics and assess differences between white and red quinoa seeds considering their flour nutritional attributes (proximal composition, fatty acids, essential amino acids profiles, minerals and antioxidant activity) and structural characteristics of the main polymers: proteins and starch (using electrophoresis, FTIR and DSC).

## 2. Materials and Methods

### 2.1. Materials

Germination of quinoa seeds. Commercial quinoa seeds used were the Royal white and Royal red quinoa purchased in a naturist health food store (La Plata, Argentina). The seeds were stored in a sealed container at 4 °C before using for analysis.

Quinoa seeds were germinated according to Guardianelli et al. [14] with slight modifications, the germination temperature being 20 °C and the germination times 0 (used as germination control sample), 18, 24 and 48 h. The seeds were then dried at 40 °C in a forced convection stove (Sanjor SL60SF, San Andrés, Buenos Aires, Argentina) to stop enzyme activity. Three independent germination tests were carried out. Finally, the seeds were ground, sieved (500 μm) and stored at −20 °C until future use.

The percentage of germination (%G) was calculated using the Equation (1):%G = germinated seeds ∗ 100/total seeds(1)

Non germinated samples were designed as: WQ for white quinoa and RQ for red quinoa. Germinated samples were: WQG18, WQG24, WQG48 for white quinoa germinated for 18, 24 and 48 h, respectively; RQG18, RQG24, RQG48 correspond to red quinoa germinated for 18, 24 and 48 h, respectively.

### 2.2. Water Activity of Quinoa Flours

Water availability was measured in a Meter AquaLab (series 4 TEV-Decagon Devices Inc., Washington, DC, USA) and assays were performed at least by triplicate.

### 2.3. Chemical Analysis of Quinoa Flours

#### 2.3.1. Proximal Composition

Moisture was determined by drying flours at 105 °C to constant weight. Content of ash was determined by calcination at 550 °C. Lipids and total protein were determined by Kjeldhal and Soxhlet methods, respectively [15]. The amount of total dietary fiber was determined according to AACC method (2000) using the enzyme-gravimetric assay (Megazyme kit).

#### 2.3.2. Non-Fiber Carbohydrates

Soluble sugars (glucose, fructose, and sucrose) and starch were quantified by High Performance Liquid Chromatography (HPLC). Firstly, from the defatted flour, the soluble sugars were extracted with Milli-Q water according to Eliasson [16]. Large soluble molecules (polysaccharides and proteins) were removed by co-precipitation with Carrez I and Carrez II. The mixture was then heated (70 °C, 600 rpm) for 30 min in a thermomixer (DLAB, Riverside, CA, USA). Subsequently, it was brought to the appropriate volume with acetonitrile and centrifuged for 10 min at 580 g.

Starch was determined in y measuring the amount of glucose by HPLC that was obtained after total hydrolysis of starch with HCl 16.7% *v*/*v* solution at reflux during 2 h. Solution was cooled at room temperature, neutralized to pH 8 and brought to an adequate final volume with acetonitrile and then was centrifuged for 10 min at 580 g.

The aqueous extracts obtained were filtered (0.45 µm) before injecting them into the Waters 1525 chromatograph (Waters Corporation, Milford, MA, USA) equipped with a Waters 2414 refractive index detector. Twenty μL of sample were injected and the sugars were separated (flow rate = 1.0 mL/min) in an isocratic form with the mobile phase (acetonitrile:water; 75:25) containing 0.2% *v*/*v* of ethylenediamine. A C18-amide column (longitude: 150 mm, internal diameter: 4.5 mm, particle size: 3.5 μm) (Thermo Fisher Scientific Inc., Waltham, MA, USA) was used. The column was kept at 30 °C, the eluent from the column entered the refractive index detector with a sensitivity of 16 [17].

Solutions of increasing concentrations (from 0.05 to 13 mg/mL) of the standard sugars glucose, fructose and sucrose (Sigma Aldrich, St. Louis, MO, USA) prepared in mobile phase were used. The peak areas and retention times were determined using the PeakFit software (version 4.12 for Windows, SPSS Inc., Chicago, IL, USA). The areas of the peaks of the sugars present in the samples were calculated and they were identified by comparison with the retention times of the standard solutions. Samples extractions were prepared in duplicate.

### 2.4. Total Phenols Content and Antioxidant Activity

#### 2.4.1. Extraction Process

Quinoa flour was extracted using acetone and acetone-water solutions according to Guardianelli et al. [14].

#### 2.4.2. Total Polyphenols

The total content of soluble phenolic compounds was determined by the Folin-Ciocalteu method modified by Asami et al. [18]. Results were expressed as milligram of gallic acid equivalents (GAE). Determinations were performed by triplicate.

#### 2.4.3. Antioxidant Activity

##### Ferric ion Reducing Antioxidant Power Assay (FRAP)

This assay was determined according to Benzie & Strain [19]. A calibration curve containing different concentrations (0–99 μg/mL; r > 0.999) of Trolox (6-hydroxy-2,5,7,8-tetramethylchroman-2-carboxylic acid) was used. FRAP values are presented as μg of Trolox per gram of flour. All determinations were performed in triplicate.

##### Free Radical Scavenger Activity on 2,2-Diphenyl-1-Picrylhydrazyl (DPPH)

The methodology of Hasperué et al. [20] was applied. Equal volumes of the extract obtained previously were mixed with the DPPH reagent, left to react for 90 min in the dark and the absorbance at 515 nm was determined. The EC50 (half maximal effective concentration) is defined as the concentration of flour required that causes a 50% decrease in the initial absorbance. Determinations were performed in triplicate.

### 2.5. Mineral Composition

Mineralization of quinoa flours and determination of the mineral content was carried out according to Guardianelli et al. [14]. Calcium, copper, iron, magnesium, manganese, phosphorus, potassium, sodium, and zinc contents were determined.

### 2.6. Fatty Acid Profile

The Fölch method was used to extract the lipids and subsequently derivatize them using a concentrated solution of HCl in methanol (5% *v*/*v*). The tubes were sealed, shaken vigorously, and then placed in boiling water for 10 min. Then, they were cooled, 2 mL of milli-Q distilled water and 1 mL of hexane were added. The tubes were capped, shaken and centrifuged (1333× *g*, 15 min) until both layers were clear. The upper phase containing the methyl esters was filtered (0.45 um). Finally, the methyl esters were injected and analyzed in an Agilent Technologies 7890A gas chromatograph (Agilent Technologies; Santa Clara, CA, USA), equipped with a Supelco DB 23 capillary column (30 m, 250 lm i.d., 0.25 mm) and with a flame ionization detector (FID). The fatty acids were identified by comparing with the retention times of Supelco 37-Component FAME Mix fatty acid methyl ester standards (Sigma-Aldrich, St. Louis, MO, USA). The area of the different peaks was analyzed using PeakFit software (version 4.12 for Windows, SPSS Inc., Chicago, IL, USA) and results were expressed as g of fatty acid per 100 g of lipid. Assays were performed by duplicate.

### 2.7. Amino Acid Composition

The amino acid profile was determined on defatted flours that were previously hydrolyzed with 6 M HCl (2 mg of protein/1.5 mL of acid solution) at 110 °C for 24 h in a reducing atmosphere. Amino acids were derivatized (50 °C, 50 min) with diethyl ethoxymethylenmalonate using D, L-α-aminobutyric acid as an internal standard; and determined by RP-HPLC (reversed-phase high performance liquid chromatography) according to Cian et al. [21]. An Agilent Model 1100 Series HPLC with VW variable wavelength detector (Santa Clara, CA, USA) equipped with a 300 mm × 3.9 mm internal diameter reverse phase column (Novapack C18, 4 m; Waters) maintained at 18 °C was used. A binary gradient at a flow rate of 0.9 mL/min was used for elution. The solvents used were sodium acetate (25 mM) containing sodium azide (0.02% *w*/*v*) pH 6.0 (A) and acetonitrile (B). Elution was carried out as follows: 0 to 3 min with linear gradient from A/B (91/9) to A/B (86/14); from 3 to 13 min with A/B (86/14); from 13 to 30 min with A/B (86/14) to A/B (69/31) and until completing 35 min with A/B (69/31). Eluted amino acids were detected at 280 nm. Tryptophan was determined by RP-HPLC after alkaline hydrolysis according to the method of Yust et al. [22]. The determinations were performed in duplicate and expressed as grams of amino acid per 100 g of protein.

### 2.8. Structural and Thermal Characterization of Protein and Starch

#### 2.8.1. Sodium Dodecyl Sulfate-Polyacrylamide Gel Electrophoresis (SDS-PAGE)

Forty mg of defatted flour were suspended in 1.5 mL of an acetone:water solution (50:50% *v*/*v*) for 40 min at 25 °C and 650 rpm. Subsequently, the mixture was centrifuged (6100× *g* for 10 min at 25 °C) and the precipitate was suspended in different buffers: (A) 0.086 M TRIS base, 0.090 M glycine and 0.003 M EDTA-pH 8 as extraction buffer (EB), (B) EB with 2% SDS and, (C) solution B with 0.5% DTT. The three extracts were shaken for 40 min (650 rpm at 60 °C) and centrifuged at 16,100 g for 10 min. Ten μL of protein extracts were mixed with sample buffer (0.5 M Tris-Base, 50% glycerol, 0.4% SDS, 5% 2-mercaptoethanol and 0.05% bromophenol blue) and next it was heated at 100 °C for 2 min. Electrophoresis was carried out following the same procedure of Guardianelli et al. [14].

The protein standards used were: phosphorylase b (97 kDa), bovine serum albumin (66 kDa), ovalbumin (45 kDa), carbonic anhydrase (30 kDa), trypsin inhibitor (20.1 kDa), and α-lactalbumin (14.4 kDa).

#### 2.8.2. Fourier-Transform Infrared Spectroscopy (FTIR)

FTIR spectra of both germinated and ungerminated quinoa flours were performed. The finely ground sample was placed in the sample holder of a Thermo Nicolet iS10 ATR-FTIR spectrometer (Thermo Scientific, Waltham, MA, USA). Thirty-two scans were taken with a spectral resolution of 4 cm^−1^ in the range 4000–500 cm^−1^, using the OMNIC software (version 8.3, Thermo Scientific, Waltham, MA, USA). Five spectra of each sample were performed.

Inverted second derivative spectra were used to estimate the number and position of the individual elements that make up the Amide I band (1580–1710 cm^−1^) related to protein secondary structure and the bands between 995 and 1150 cm^−1^ corresponding to the starch organization. This information was taken into account to adjust the bands of Amide I in protein spectra with Gaussian * Lorenzian band profiles, and in starch it was adjusted with Gaussian band profiles, using PeakFit software (version 4.12 for Windows, SPSS Inc., Chicago, IL, USA). The assignment of the secondary protein structures to the main frequencies of Amide I was carried out according to Gerbino et al. [23], while those of the starch structure according to Monroy et al. [24].

#### 2.8.3. Differential Scanning Calorimetry (DSC)

The thermal properties of quinoa flours were determined on their aqueous suspensions (40% *w*/*v*) according Menegassi [25] modified by Salinas et al. [17] by differential scanning calorimetry, using a Q100 equipment (TA Instruments, New Castle, DE, USA). Protein denaturation and/or starch gelatinization were characterized by different temperatures: onset (T0), peak (Tp) and final (Tf). The enthalpy associated with the endothermic process (ΔH) was determined between T0 and Tf. Assays were analyzed by duplicate.

### 2.9. Statistical Analysis

Variations between different treatments were studied by one-way analysis of variance (ANOVA) using INFOSTAT software. LSD Fisher test at *p* < 0.05 was considered for significant differences between mean values. Principal component analysis (PCA) was carried out to determine the correlation between quinoa varieties and the effect of quinoa germination using Infostat software [26].

## 3. Results and Discussion

As expected, the germination percentage of white quinoa seeds increased significantly over time (WQG18 = 53%, WQG24 = 72%, and WQG48 = 85%), with mean sprout length being 0.6, 0.9 and 1.3 cm for WQG18, WQG24 and WQG48, respectively. Red quinoa seeds followed a similar trend with values of 50, 69 and 80% for RQG18, RQG24 and RQG48, respectively; and the sprouts presented a shoot length of 0.5 (RQG18), 0.8 (RQG24) and 1.1 cm (RQG48).

### 3.1. Water Availability and Chemical Composition of Quinoa Flours

The availability of water measured as the water activity (aw) in the white (WQ) and red (RQ) quinoa flours obtained from the non-germinated seeds was 0.6287 and 0.4683, respectively. On the other hand, the germinated white quinoa flours presented different aw values: 0.3450 (WQG18), 0.3159 (WQG24) and 0.4357 (WQG48), while the water activity values for the germinated red quinoa flours were: 0.2775 (RQG18), 0.5228 (RQG24) and 0.3786 (RQG48). Although the value of aw was variable, in all samples did not exceed the value of 0.7, which is the reason for considering flours safe ingredients for foods from the point of view of fungal alteration [27].

The composition of both ungerminated and germinated quinoa varieties flours is shown in Figure 1. The moisture of germinated white quinoa flours was lower than that of WQ and no differences were observed in the ash content (Figure 1A). The germinated flours presented a significant increase in the content of lipids and total dietary fiber (TFD), being these components more abundant in the first stages of germination. Furthermore, protein content increased with the germination time (24 and 48 h).

For red quinoa flours, the moisture and protein content at high germination times of samples RQG24 and RQG48 were lower than in RQG18 and RQ flours. The amount of ash had a slight increase with germination. Lipid and total dietary fiber content increased after 24 h of germination (Figure 1B).

Table 1 shows the available carbohydrate content of all quinoa flours. In white quinoa flours, it was observed that germination increased the content of fructose, glucose and sucrose, while the starch content practically did not change. A similar trend was observed in red quinoa flours for fructose, glucose and sucrose content, however, starch content decreased with the germination time (≥24 h). The significant decrease in the starch content for red quinoa (RQG48) could be due to the fact that starch has a structural arrangement that is more accessible to hydrolytic enzymes.

Padmashree et al. [11] and Antezana et al. [28] studied the chemical composition of red and white quinoa seeds after 48 h of germination at room temperature, reporting differences in behavior between varieties. However, in this work we studied the changes in composition at different germination times, finding different trends (Figure 1) according to the quinoa variety.

Regarding the increase in TDF Padmashree et al. [11] observed a similar trend in both quinoa varieties (48 h). This increase could be attributed to the fact that during germination a new plant is being formed and consequently its cell wall which contains a high proportion of insoluble fiber. For white quinoa, we observed a TDF increase at 18 h with a subsequent decrease at 24 h, value that remained constant at 48 h. For red quinoa the TDF increase was progressive over time. These differences in TDF behavior suggest that fiber content during germination depends on the kind of variety.

With respect to lipids, in our case both varieties had different transformation kinetics. For white quinoa at 18 h it increased, remained constant at 24 h and then decreased at 48 h; while for red quinoa up to 18 h the lipid level was constant, increasing significantly at 24 h and then a decrease at 48 h was observed. These differences suggest an accelerated first stage of anabolism for white quinoa respect to red quinoa, and after 24 h lipid catabolism predominates in both varieties. The behavior after 48 h of germination was similar to that observed by Padmashree et al. [11] and Antezana et al. [28]. Part of the lipids is degraded to obtain energy for the development of the new plant.

In our work, up to 18 h the protein content was constant for both quinoas, while after this time a differential behavior was observed between white and red quinoa. For white quinoa, proteins, as in the case of Padmashree et al. [11], increased, and in our case this increase was already registered after 24 h; this effect was attributed to the protein synthesis. For red quinoa, in the same manner as Antezana et al. [28], the protein level slightly decreased due to its degradation for the formation of other components.

In summary, despite having germinated under the same conditions (time and temperature), white quinoa had a rapid response to germination, increasing at 18 h the synthesis of proteins, perhaps forming the enzymes necessary for plant growth; while red quinoa also showed a delay in the variation of lipids, increasing only the level of fiber in a sustained way until 48 h.

### 3.2. Changes in Total Phenolic Compounds and Antioxidant Activity

Table 2 shows the content of total phenolic compounds (TFC). The red quinoa presented higher TFC than the white sample. In the case of white quinoa flour, the TFC content increased significantly with germination time, presenting WQG48 flour the highest value. In the case of red quinoa flour, a very similar behavior was observed.

In addition, the antioxidant activity (AA) measured by two complementary methods (DPPH and FRAP) is shown in Table 2. Regarding white quinoa flours, WQG48 presented the highest antioxidant activity measured by FRAP, and the EC50 was lower in all the germinated flours (WQG) regardless of the germination time. In red quinoa a different behavior was observed; the antioxidant activity decreased with germination, being lower than in RQ in both methods (Table 2).

Various authors found that the content of phenolic compounds and AA in white and red quinoa flour increased with germination [10,11,13]. Similar trend in both parameters was found in this work for white quinoa but different for the red variety, as it can be deduced from values of Table 2.

### 3.3. Minerals

Certain minerals (Ca, Na, Fe and Zn) present in the flours obtained from white quinoa seeds, germinated or ungerminated, can be observed in Table 2. In the case of white quinoa, the calcium content was the highest, followed by sodium, iron and zinc. A higher content of Ca and Na in germinated white quinoa flours (Table 2) was observed, while the content of Fe and Zn showed a slight decrease at 48 h of germination. Despite the increase in Na, the content of this mineral in 100 g of flour is much lower than the recommended daily intake (2400 mg/day) according to the Argentine Food Code (2022).

On the other hand, in RQ the highest content was presented by sodium, followed by calcium, iron and zinc. The germinated red quinoa flours (Table 2) presented an increase in the content of Ca and Fe (RQG18 and RQG48); while Zn practically did not change, meanwhile the content of Na decreased as a result of germination, being minimum in RQG24.

Other authors found an increase in the content of Ca, Fe, Zn in white quinoa sprouts at 48 h [29] and 72 h [30] compared to that without germination, while Bhinder et al. [31] found in germinated white quinoa, an increase in content of Fe, a decrease in K, Mg and Zn, and no differences in Ca, Cu and Mn. On the other hand, in germinated black quinoa no changes were observed in the content of Ca, Mn, Cu and Fe, but there was a decrease in K, Mg and Zn [31].

The different trends found in the minerals allow us to conclude that the mineral content depends on the initial quinoa variety and the germination conditions. Changes in these micronutrients could be due to the hydrolysis of complex organic compounds that could release minerals during germination [32] and they could act as enzymatic cofactors which would make them vary according to the stage of the process.

### 3.4. Fatty Acids

Table 2 also shows the content of unsaturated fatty acids present in quinoa flour. In ungerminated white quinoa (WQ) it was observed that linoleic acid (18:2) was the majority (49%), followed by oleic acid (18:1), α-linolenic acid (18:3) and eicosenoic acid (20:1). During germination the oleic acid content (18:1) decreased. In turn, the essential fatty acids linoleic (18:2-ω6) and α-linolenic (18:3-ω3) increased with germination, presenting the highest values in the WQG48 flour. This behavior could be attributed to the action of desaturase enzymes that are activated during the germination process that convert oleic acid into linoleic and α-linolenic acid [33]. 

At the same time, although both fatty acids are essential, excessive intake of ω6 present in foods consumed to a greater extent (legumes, oils, avocado, among others) together with lower consumption of foods with ω3 (seeds, some fish, nuts, among others) generates an imbalance in the optimal relationship that can cause a deterioration in health. This is why the World Health Organization (WHO) establishes that the maximum value of the ω6/ω3 ratio should be 10:1 [34]. The data obtained shows that although the content of total unsaturated fatty acids did not change with germination, an improvement in quality was observed due to the fact that ω6 and ω3 increased, keeping the ω6/ω3 ratio constant, except in WQG48 (5.3), where the relationship was minimum (Table 2). Thus, WQG48 flour would be an attractive ingredient in Western diets since it improves the balance between both fatty acids.

For non-germinated red quinoa (RQ) a greater amount of linoleic acid, respect to white quinoa (WQ) was detected; followed by oleic and eicosenoic acids, but in a different proportion (Table 2). It was observed that, during germination, the content of oleic and α-linolenic acid increased, while linoleic acid decreased. In addition, germination generated a decrease in the ω6/ω3 ratio, presenting the lowest value for RQG24.

Park and Morita [35] found from the analysis of fatty acids present in white quinoa seeds, that the ratio ω6/ω3 for the first times of germination (24 and 48 h at 30 °C) increased compared to that without germination, while the value after 72 h of germination was the lowest one.

### 3.5. Amino Acids Composition

Table 2 shows the values of essential amino acids of proteins of all quinoa samples. For white quinoa, the levels of lysine, isoleucine, tyrosine + phenylalanine and threonine decreased; the content of valine, leucine, methionine and tryptophan practically did not change, while the content of histidine increased during germination. In general, it was observed that the total content of essential amino acids decreased throughout germination, however, in all cases, the content of each essential amino acid was above the FAO requirements for adults [36], with methionine being the limiting amino acid.

For red quinoa (Table 2) it was observed that there were practically no variations in the level of essential amino acids with germination time, however, the content of total essential amino acids was higher in the RQG24 flour.

Bhatal et al. [37] found in germinated white quinoa seeds (4–5 h), that the content of lysine and tryptophan in the germinated samples was lower than in ungerminated quinoa; no differences were observed in methionine. On the other hand, Fouad and Rehab [7] studied the effect of lentil germination time on the amino acid profile, finding a decrease in Ile, Thr, Met and an increase in His, as in our case. These authors attributed this behavior to the mobilization of protein reserves in the cotyledons, together with the synthesis of new proteins, necessary for shoot growth. Furthermore, they pointed out that the amino acids produced by the hydrolysis of storing proteins are not only used to synthesize new components, but also can be used as an energy source, especially in the early stages of germination [7].

In summary, germination for 48 h increased the content of TDF and lipids, without modifying the protein content. WQG48 presented the highest content of linoleic acid (18:2-ω6) and α-linolenic acid (18:3-ω3) and the best ω6/ω3 ratio; and also presented the highest content of phenolic compounds and antioxidant activity. On the other hand, for the red quinoa high TDF, protein and lipid contents with a high percentage of essential amino acids (Lys, His and Met) was obtained after 24 h of germination (RQG24). High amounts of α-linolenic and oleic fatty acids were observed, and this sample presented the highest amount of phenolic compounds and the lowest sodium content.

Therefore, based on the results described, white quinoa flour germinated for 48 h (WQG48) and red quinoa germinated for 24 h (RQG24) were selected for structure and thermal analysis due to the fact that they are the ones that presented the best nutritional characteristics (except amino acid composition).

### 3.6. Protein Fractions

Figure 2 shows the SDS-PAGE profiles of white and red quinoa proteins extracted with different buffers. The extraction buffer of pH 8 (Figure 2A,B-line B), a buffer that dissociates electrostatic bonds, allowed in both varieties of quinoa (white and red) to extract mainly proteins of molecular weight 66, 58, 17 kDa and to a minor extent proteins of 56, 42, 35, 26 and 20 kDa. A very intense band of molecular weight less than 14.4 kDa was observed, which moved practically with the running front. After germination, a high amount of proteins of 66, 58 and 35 kDa was observed for red quinoa, and a great proportion of these proteins were released and solubilized in this pH 8 buffer, suggesting changes in protein structure due to germination process.

The addition of SDS to the extraction buffer (line B + SDS), which is a denaturing agent, significantly increased protein extraction in both type of quinoas (Figure 2A,B, WQ and RQ). SDS solubilized high molecular weight aggregates (proteins that did not enter the gel) in all samples. The most intense bands were those of 66, 63 and 58 kDa, suggesting that these proteins are forming protein aggregates linked by non-covalent bonds: hydrophobic and hydrogen bonds. The 63 kDa band is present only in the white quinoa extract (WQ-B + SDS) and it was absent in the WQ-B extract, suggesting that the pH 8-buffer was not able to dissociate electrostatic bonds between proteins; but this protein was solubilized with the SDS denaturing agent. The absence of the 63 kDa protein in the red quinoa extract would indicate that there are differences in the way in which proteins are interacting each other in each variety, that is, they could have a different structure despite having the same molecular weight (MW). On the other hand, it was possible to solubilize or dissociate proteins with a molecular weight of 26, 20, 19, 17 kDa and largely the band of 15 kDa; all of them were linked between them or to other proteins of major size (>50 kDa) by non-covalent bonds. The same proteins were extracted from germinated samples.

Finally, the extraction buffer with SDS with the addition of DTT (Figure 2A,B, line B + SDS + DTT) also produced dissociation of proteins linked by covalent disulfide bonds (-S-S-). Soluble high molecular weight aggregates are absent due to they were dissociated into small molecules. The most intense bands, present in all profiles, were those of 35 and 26 kDa; and to a lesser extent those of 42 and 20 kDa. The high intensity of these bands is in concordance with the dissociation of the 11S globulin subunit (55–62 kDa) due to the fac that they are bound by disulfide bonds [38]. In addition, proteins with MW < 14.4 kDa were assigned by Janssen et al. [38] to the 2S albumin fraction. It is evident that the 35 kDa protein was linked by S-S bonds to other proteins, it could be forming the 66 and 58 kDa proteins whose intensity decreased significantly with respect to the B+SDS line. The main differences between varieties were a greater proportion of the 63 kDa band in white quinoa respect to red quinoa, and a higher proportion of 35 kDa protein (more intense band) in red quinoa in comparison to the white variety.

From this analysis, it could be inferred that the protein structure of white quinoa is different from that of red quinoa; despite having the same fractions or polypeptides, they are stabilized into larger proteins or aggregates by different types of linkages. On the other hand, the effect of germination on protein dissociation was mostly evidenced in red quinoa.

### 3.7. Secondary Structure of Proteins and Starch Organization

Results of the deconvolution of Amide I of FTIR spectra of the different quinoa flours are shown in Table 3. The ungerminated white quinoa sample (WQ) presented a high proportion of parallel β-sheet structure, mostly intermolecular, and a low percentage of α-helix and β-turns. Germination (WQG48) slightly decreased the intermolecular parallel β-sheet and increased the α-helix, without significant changes in the rest of the secondary structures. These results suggests that germination in white quinoa produced changes in protein structure favoring a more folded or compact structure in comparison to non-germinated quinoa. Change in starch organization as consequence of germination was studied by the analysis of the 1045/1022 and 1022/995 areas ratios (Table 3). The higher values of 1045/1022 ratio are associated with a higher order of the crystalline regions, while the ratio of the bands 1022/995 represents the state of organization of the double helices located within the crystals [24]. No differences in 1045/1022 and 1022/995 ratios for white quinoa were found, indicating that no differences in the structure of the starch between WQ and WQG48 were detected.

On the other hand, after 24 h of germination (RQG24), proteins of red quinoa presented a more unfolded secondary structure than in RQ, evidenced by an increase in intramolecular parallel β-sheets and a decrease in α-helix (Table 3). About the structure of starch in red quinoa (Table 3), the 1045/1022 ratio increased as a result of germination, while 1022/995 decreased, indicating a decrease in the amorphous regions of the starch. Xing et al. [39] found similar results in the analysis by FTIR of structural properties of starch isolated from ungerminated white and red quinoa and germinated for 24 and 48 h. From the intensity 1045 cm^−1^/1022 cm^−1^ ratio they estimated the amount of crystalline starch. They found values of this ratio ranged between 0.580 and 0.654, with the lower values belonging to the germinated samples. This behavior suggests that this treatment induce alteration in the ordered structure of the starch granules.

### 3.8. Thermal Properties

Figure 3 shows the thermograms obtained by DSC of the dispersions of quinoa flours, both germinated and non-germinated. An endothermic peak was observed in white quinoa samples between 65–66 °C (Figure 3A), which corresponds to starch gelatinization [39] together with protein denaturation [40]. The enthalpy change (ΔH) in WQ was 53 J/g. The germination process did not modify the endothermic peak temperature nor the enthalpy variation value.

The amount of starch was not modified with germination (Table 1) and no changes were found in the crystalline and amorphous regions by means of FTIR (Table 3). Considering that the enthalpy value has a contribution of both protein denaturation and starch gelatinization, and that was not modified by germination, this behavior could be explained by the high contribution of starch to this thermal property.

In addition, Figure 3B also shows the thermograms corresponding to the suspensions of red quinoa. In both samples an endothermic peak between 72–73 °C was observed. The enthalpy value of RQ was 63.3 J/g, greater than in WQ. Unlike what was observed with white quinoa, germination produced a significant decrease in enthalpy variation that can be associated with the lower content of starch after germination, as observed in Table 1, and the high crystallinity observed by the higher 1045/1022 ratio in Table 3. Likewise, the decrease in protein content with germination could have been contributed to the less energy required in endothermic processes.

However, unlike our results, Xing et al. [39] studied the thermal behavior of starch suspensions (50% *w*/*v*) extracted from germinated white and red quinoa at 25 °C for 24 and 48 h, finding that the enthalpy variation of gelatinization increased with germination process. They attributed this effect to the fact that amylases acted on the amorphous regions of the starch, which increases the ordered double helix structure, requiring more energy for the gelatinization of the germinated quinoa starch. Difference with our results may be due to the fact that in the non-germinated flours the starch is in native state and is bound to other components, and partially hydrolyzed in the germinated ones; while the starch isolated by these authors has undergone structural changes as a consequence of the extraction process.

Jimenez et al. [41] studied thermal properties on germinated and non-germinated white quinoa flours. They found similar behavior that red quinoa studied in this work. Values of ΔH decreased in suspensions of germinated flour (24 °C for 24 h) compared to non-germinated quinoa, attributing this behavior to the decrease in starch content after germination, in concordance with our results shown in Table 1.

### 3.9. Principal Component Analysis (PCA)

Figure 4 shows the Principal Component Analysis (PCA). When analyzing principal component 1 (PC1), which describes 46.9% of the variance, it was observed that this component allows separating the two varieties of quinoa: in one side white quinoa (towards the positive values of PC1) where the starch and protein contents and antioxidant activity have a greater incidence. On the other hand, for red quinoa (towards the negative values of PC1), the parameters that have the greatest influence are the content and type of lipids and dietary fiber; that is, those variables referred to the chemical aspect (composition) have a greater weight on this component.

On the other hand, with regard to component 2 (PC2), which explains 25.4% of the variance, it can be seen that this component shows the effect of germination on quinoa seeds. Quinoa flours without germination (negative axis of PC2) were mostly affected by moisture, lysine content and enthalpy variation (ΔH) parameters. Meanwhile, quinoa flours germinated at ≥24 h (positive axis of PC2) had a major impact of sugar content, starch crystallinity and a high proportion of secondary structure of the protein in the form of α-helix. This trend is consistent with the higher enthalpy value in the ungerminated flours, since in these flours there was a higher starch crystalline structure and a more compact protein structure, which would require a higher enthalpy to denature proteins and gelatinize the starch. By this component it can be observed that the greatest influence of the variables is related to the physicochemical characteristics of the flours.

From this analysis, it can be inferred that the greatest difference observed in the samples is due to the variety of quinoa (white or red); since, although both quinoas were modified during germination, they were manipulated in a similar way. Therefore, the differences observed between them are mainly due to the type of variety.

A comprehensive analysis of the results obtained with the different trials gives an idea of the effect of germination on the quinoa variety and the consequences on the quality of a food.

In the case of red quinoa, 24 h of germination (RQG24) resulted in flour with 13% more of lipids and 7% more of total dietary fiber (TDF), in comparison to the ungerminated red quinoa. After germination, protein underwent structural changes with higher unfolding and starch presented a lower proportion of amorphous region, both evidenced by FTIR. These changes coincided with a decrease in the enthalpy of the endothermic processes occurring simultaneously and in the same temperature range (60–90 °C); which includes starch gelatinization and/or protein denaturation. In contrast, white quinoa germinated for 48 h (WQG48) presented higher TDF (48% more), protein (8% more) and lipid (11% more) contents compared to ungerminated white quinoa, with no changes in the starch content. For this flour, protein presented a high compact structure, and the amorphous and crystalline regions of the starch were not affected by germination. This behavior is in agreement with the nonvariation of the gelatinization/denaturation enthalpy. From the nutritional point of view, the effect of germination was more positive in white quinoa; however, the components (proteins and starch) of red quinoa presented greater structural changes. The quinoa flours are highly used as ingredients in baked products, mainly breads. The changes in composition and structure of biopolymers of quinoa flours, due to germination, would affect the gluten network with consequences on bread quality.

Very high dietary fiber content, as in the case of WQG48, could interfere with the gluten network and result in breads of smaller volume. In addition, a higher proportion of native or compacted polymeric proteins absorb less water and would form a less extensible dough that is able to form loaves of smaller volume and compact crumbs. Gluten proteins (gliadins and glutenins) are responsible for conferring optimal viscoelastic properties to the dough; a higher proportion of polymeric proteins in the unfolded state and starch in the crystalline state, as in the case of RQG24, could increase water absorption by the gluten, leading to more extensible dough, resulting in more aerated crumbs and greater bread volume.

## 4. Conclusions

The germination process of both quinoa varieties led to changes in their composition and structure. In the case of white quinoa, the WQG48 flour presented a higher content of total dietary fiber and lipids, with increased levels of the unsaturated fatty acids Linoleic (18:2-ω6) and α-Linolenic (18:3-ω3) leading a better ratio ω6/ω3. Likewise, the content of phenolic compounds and antioxidant activity and histidine increased in this flour. During germination, hydrolysis of starch and proteins occurred, the secondary structure of proteins in WQG48 was the most compact (by FTIR) and thermally stable (by DSC). On the other hand, for red quinoa, the RQG24 flour had a higher protein, lipid and total dietary fiber content. The nutritional profile of the flour was improved with the increase of essential amino acids (His and Met) and α-linolenic and oleic fatty acids. A higher content of phenolic compounds and a lower level of sodium contributed to this profile. Germination modified the secondary structure of the proteins making them more unfolded as well as contributing to the crystallinity of the starch (FTIR). The lower starch content of QRG24 was evidenced by the lower variation of enthalpy of gelatinization (DSC). In conclusion, if it is desired to use these flours in wheat flour-based breads for nutritional improvement, it should be taken into account that a high content of fiber and compacted proteins (germinated white quinoa) could change the gluten network of the dough, generating smaller breads. On the contrary, a higher proportion of unfolded polymeric proteins and a more crystalline starch (germinated red quinoa) would generate more aerated crumbs and a larger loaf volume. Differences in both kind of quinoas will influence physicochemical properties of flours and therefore the technological quality of the foods prepared with these nutritional improved seeds.

## Figures and Tables

**Figure 1 foods-11-03272-f001:**
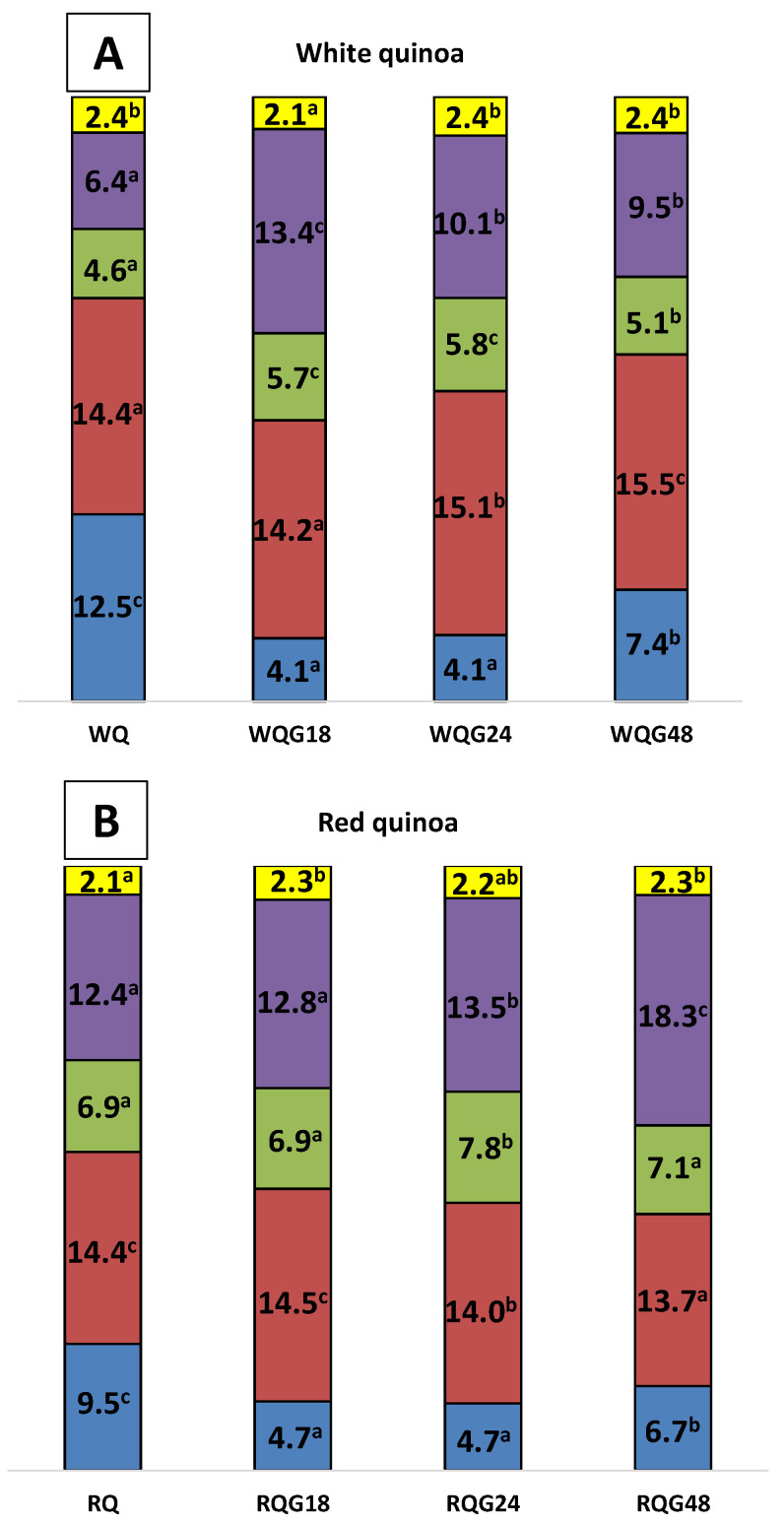
Proximal composition of flours. Percentage of moisture (blue), proteins (red), lipids (green), total dietary fiber (violet) and ashes (yellow). (**A**) White quinoa flour without germination (WQ) and germinated 18 h (WQG18), 24 h (WQG24) and 48 h (WQG48). (**B**) Ungerminated red quinoa flour (RQ) and germinated 18 h (RQG18), 24 h (RQG24) and 48 h (RQG48). Different letters indicate significant differences in the same variety of quinoa (*p* < 0.05).

**Figure 2 foods-11-03272-f002:**
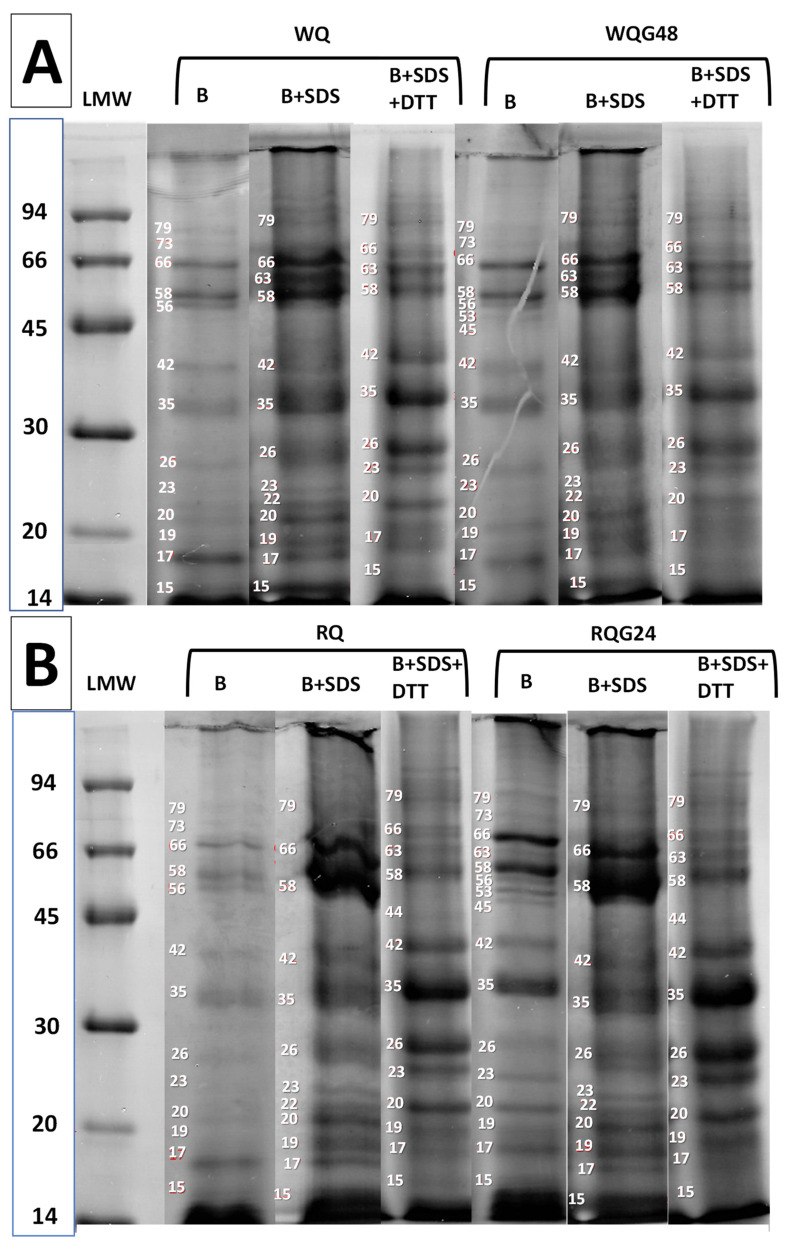
SDS-PAGE of proteins from quinoa flours: WQ: white quinoa, and WQG48: germinated 48 h (**A**). RQ: red quinoa, and RQG24: germinated 24 h (**B**). Protein extracted in buffer pH = 8 (**B**), buffer pH = 8 with 2% *w*/*v* SDS (B + SDS) and buffer pH = 8 with 2% *w*/*v* SDS and 0.5% *w*/*v* DTT (B + SDS + DTT).

**Figure 3 foods-11-03272-f003:**
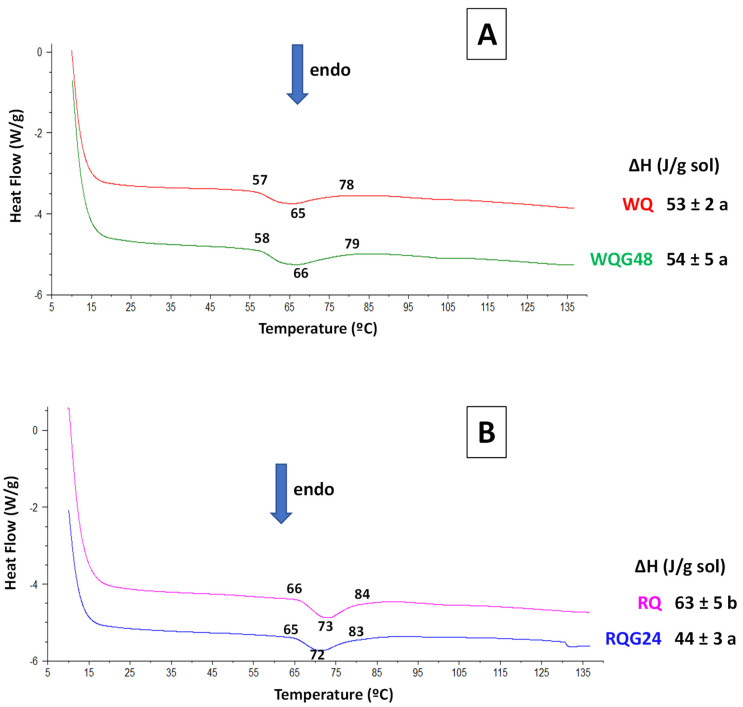
DSC thermograms of the different suspension of quinoa flours: (**A**) WQ: white quinoa, WQG48: germinated 48 h. (**B**) RQ: red quinoa, and RQG24: germinated 24 h. Values of temperatures and enthalpy change of the endothermic process (ΔH) are shown.

**Figure 4 foods-11-03272-f004:**
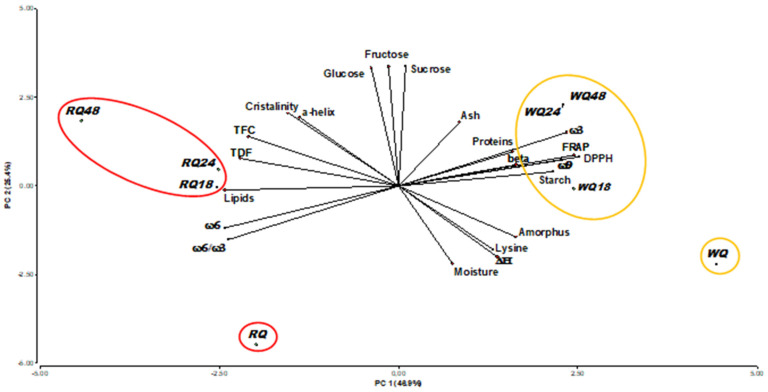
Biplot (PCA) of the correlation between quinoa varieties and quinoa germination.

**Table 1 foods-11-03272-t001:** Available carbohydrate content of ungerminated and sprouted white and red quinoa flours.

Flours	Fructose (%)	Glucose (%)	Sucrose (%)	Starch (%)
WQ	0.22 ± 0.04 ^a^	0.91 ± 0.01 ^a^	0.80 ± 0.10 ^a^	57.72 ± 0.78 ^a^
WQG18	0.34 ± 0.03 ^b^	1.49 ± 0.01 ^b^	0.92 ± 0.06 ^ab^	57.72 ± 0.82 ^a^
WQG24	0.44 ± 0.02 ^c^	1.53 ± 0.02 ^bc^	0.96 ± 0.12 ^ab^	59.63 ± 0.53 ^b^
WQG48	0.53 ± 0.02 ^d^	1.58 ± 0.01 ^c^	1.00 ± 0.08 ^b^	57.06 ± 0.84 ^a^
RQ	0.18 ± 0.01 ^a^	0.80 ± 0.00 ^a^	0.61 ± 0.10 ^a^	53.11 ± 1.71 ^b^
RQG18	0.30 ± 0.01 ^b^	1.46 ± 0.01 ^b^	0.89 ± 0.10 ^b^	56.2 ± 1.5 ^c^
RQG24	0.37 ± 0.01 ^c^	1.51 ± 0.07 ^b^	0.98 ± 0.05 ^b^	54.93 ± 0.91 ^b^
RQG48	0.48 ± 0.01 ^d^	1.52 ± 0.01 ^b^	0.96 ± 0.07 ^b^	48.97 ± 0.98 ^a^

Different letters in the same column for the same variety of quinoa indicate significant differences (*p* < 0.05).

**Table 2 foods-11-03272-t002:** Antioxidants, minerals and essential components of ungerminated and sprouted white and red quinoa flours.

	Total Phenolics Content—Antioxidant Activity					Minerals			
Flours	Folin (mg GAE/100 g Flour)		DPPH (EC50) (mg/µL)		FRAP (mg Trolox/100 g Flour)	Sodium (ppm)	Iron (ppm)	Zinc (ppm)	Calcium (ppm)
WQ	94.3 ± 12.4 ^a^		22.1 ± 0.2 ^b^		113.7 ± 0.8 ^a^	200 ± 4 ^a^	47.3 ± 1.3 ^b^	27.2 ± 0.1 ^b^	438 ± 25 ^a^
WQG18	111.1±3.8 ^b^		20.8 ± 0.7 ^a^		123.4 ± 0.7 ^c^	345 ± 10 ^b^	51.6 ± 1.1 ^c^	56.1 ± 0.4 ^c^	738 ± 57 ^b^
WQG24	105.6±8.5 ^b^		20.6 ± 0.0 ^a^		120.1 ± 0.5 ^b^	352 ± 17 ^b^	45.5 ± 0.9 ^b^	26.3 ± 0.5 ^b^	766 ± 53 ^b^
WQG48	121.5 ± 0.7 ^c^		19.9 ± 0.2 ^a^		127.4 ± 0.7 ^d^	367 ± 18 ^b^	39.7 ± 0.7 ^a^	24.3 ± 0.2 ^a^	729 ± 50 ^b^
RQ	113.7 ± 0.8 ^a^		7.6 ± 0.0 ^a^		72.0 ± 1.4 ^c^	914 ± 66 ^c^	33.6 ± 0.8 ^b^	21.1 ± 0.1 ^a^	413 ± 16 ^a^
RQG18	123.4± 0.7 ^c^		8.2 ± 0.0 ^b^		62.6 ± 1.7 ^a^	684± 33 ^b^	35.8 ± 0.6 ^c^	36.8 ± 0.5 ^b^	988 ± 26 ^b^
RQG24	120.1±0.5 ^b^		8.0 ± 0.1 ^b^		67.2 ± 0.9 ^b^	503 ± 24 ^a^	31.9 ± 0.4 ^a^	20.9 ± 0.4 ^a^	427 ± 10 ^a^
RQG48	127.4±0.7 ^d^		8.7 ± 0.2 ^c^		63.0 ± 1.9 ^a^	682 ± 52 ^b^	35.5 ± 0.8 ^c^	21.0 ± 0.4 ^a^	951 ± 82 ^b^
	Unsaturated fatty acids (g/100 g lipids)								
Flours	Oleic acid (18:1-ω9)		Linoleic acid (18:2-ω6)		α-Linolenic acid (18:3-ω3)	Eicosenoic acid (20:1-ω9)		ω6/ω3 ratio	
WQ	30.0 ± 0.0 ^c^		49.0 ± 0.1 ^a^		8.7 ± 0.0 ^a^	1.3 ± 0.0 ^a^		5.6 ± 0.0 ^b^	
WQG18	30.0 ± 0.1 ^c^		49.0 ± 0.0 ^a^		8.7 ± 0.1 ^a^	1.4 ± 0.1 ^a^		5.6 ± 0.1 ^b^	
WQG24	29.8 ± 0.0 ^b^		48.9 ± 0.0 ^a^		8.9 ± 0.0 ^b^	1.4 ± 0.0 ^a^		5.5 ± 0.0 ^b^	
WQG48	28.4 ± 0.1 ^a^		49.8 ± 0.2 ^b^		9.4 ± 0.0 ^c^	1.3 ± 0.0 ^a^		5.3 ± 0.0 ^a^	
RQ	27.0 ± 0.0 ^a^		53.2 ± 0.1 ^c^		6.2 ± 0.0 ^a^	1.3 ± 0.1 ^a^		8.6 ± 0.1 ^d^	
RQG18	28.0 ± 0.1 ^c^		52.2 ± 0.0 ^a^		6.6 ± 0.0 ^b^	1.3 ± 0.1 ^a^		7.9 ± 0.0 ^c^	
RQG24	27.8 ±0.0 ^bc^		52.1 ± 0.0 ^a^		6.8 ± 0.0 ^d^	1.4 ± 0.0 ^a^		7.6 ± 0.0 ^a^	
RQG48	27.6 ± 0.2 ^b^		52.3 ± 0.1 ^b^		6.7 ± 0.0 ^c^	1.4 ± 0.0 ^a^		7.8 ± 0.0 ^b^	
	Essential amino acids (mg/100 g de proteins)								
Flours	Lysine (4.5 *)	Histidine (1.5 *)	Valine (3.9 *)	Isoleucine (3.0 *)	Leucine (5.9 *)	Methionine (1.6 *)	Tyrosine + Phenylalanine (3.8 *)	Threonine (2.3 *)	Tryptophan (0.6 *)
WQ	16.6 ± 1.6 ^b^	6.4 ± 0.8 ^a^	6.5 ± 0.3 ^bc^	15.8 ± 0.2 ^b^	10.8 ± 0.8 ^a^	1.2 ± 0.6 ^ab^	10.9 ± 1.8 ^b^	8.1 ± 0.5 ^b^	1.5 ± 0.1 ^a^
WQG18	15.6 ± 1.0 ^b^	5.4 ± 0.8 ^a^	6.7 ± 0.5 ^c^	15.4 ± 0.8 ^b^	10.8 ± 0.9 ^a^	1.7 ± 0.3 ^b^	10.8 ± 0.8 ^b^	7.7 ± 0.4 ^b^	1.3 ± 0.0 ^a^
WQG24	11.9 ± 2.5 ^a^	12.0 ± 0.9 ^c^	4.4 ± 0.1 ^a^	12.2 ± 2.2 ^a^	10.8 ± 0.1 ^a^	0.9 ± 0.1 ^a^	8.6 ± 1.2 ^a^	6.1 ± 0.2 ^a^	1.5 ± 0.3 ^a^
WQG48	12.5 ± 0.9 ^a^	10.5 ± 1.9 ^b^	5.7 ± 0.4 ^b^	12.2 ± 0.5 ^a^	9.5 ± 0.1 ^a^	0.8 ± 0.1 ^a^	8.5 ± 0.5 ^a^	6.5 ± 0.3 ^a^	1.6 ± 0.2 ^a^
RQ	13.2 ± 0.6 ^a^	6.8 ± 1.5 ^a^	5.5 ± 0.3 ^a^	14.0 ± 1.2 ^a^	9.5 ± 0.3 ^ab^	1.1 ± 0.2 ^b^	9.4 ± 0.6 ^a^	6.7 ± 0.7 ^ab^	1.2 ± 0.1 ^a^
RQG18	13.3 ± 0.7 ^a^	7.0 ± 1.1 ^a^	5.7 ± 0.1 ^a^	13.9 ± 0.7 ^a^	10.5 ± 0.4 ^b^	1.3 ± 0.1 ^b^	9.3 ± 0.9 ^a^	7.4 ± 0.4 ^b^	1.4 ± 0.1 ^a^
RQG24	13.9 ± 2.1 ^a^	22.1 ±1.7 ^b^	5.1 ± 0.6 ^a^	12.1 ± 1.1 ^a^	9.3 ± 0.7 ^a^	2.5 ± 0.4 ^c^	9.1 ± 0.5 ^a^	6.5 ± 0.1 ^a^	1.7 ± 0.4 ^a^
RQG48	12.2 ± 0.2 ^a^	7.0 ± 1.4 ^a^	5.7 ± 0.5 ^a^	12.7 ± 0.1 ^a^	9.1 ± 0.3 ^a^	0.3 ± 0.1 ^a^	8.5 ± 0.9 ^a^	6.9 ± 0.2 ^ab^	1.3 ± 0.1 ^a^

* Amino acid requirements in human nutrition—FAO/OMS/UNU, 2007. Different letters in the same column for the same variety of quinoa indicate significant differences (*p* < 0.05).

**Table 3 foods-11-03272-t003:** Wavenumber, protein secondary structure percentage (Amide I) and starch structure ratio of ungerminated and sprouted white and red quinoa flours.

Secondary Structure	Wavenumber (cm^−1^)	Flours			
		WQ	WQG48	RQ	RQG24
β-sheet parallel intermolecular	1620	42 ± 2.5 ^b^	37.3 ± 2.4 ^ab^	30.3 ± 4.1 ^a^	31.1 ± 3.0 ^a^
β-sheet parallel intramolecular	1636	18.5 ± 1.8 ^a^	19.7 ± 1.9 ^ab^	20.8 ± 1.2 ^a^	26.9 ± 2.8 ^b^
Random coil	1650	21.6 ± 0.5 ^b^	17.9 ± 2.1 ^ab^	27.1 ± 0.7 ^b^	25.2 ± 3.0 ^ab^
α-helix	1661	6.7 ± 1.0 ^a^	11.3 ± 0.2 ^c^	10.7 ± 0.4 ^b^	8.8 ± 1.5 ^a^
β-turn	1674	6.4 ± 1.1 ^a^	6.1 ± 1.4 ^a^	5.7 ± 0.05 ^a^	7.0 ± 0.1 ^b^
β-sheet antiparallel intramolecular	1684	6.0 ± 0.8 ^a^	7.5 ± 0.2 ^ab^	6.6 ± 0.7 ^b^	5.1 ± 0.1 ^a^
Starch	Wavenumber (cm^−1^)				
Area ratio (crystalline)	1045/1022	0.36 ± 0.02 ^ab^	0.37 ± 0.01 ^ab^	0.35 ± 0.05 ^a^	0.45 ± 0.03 ^b^
Area ratio (amorphous)	1022/995	0.51 ± 0.04 ^ab^	0.48 ± 0.01 ^a^	0.56 ± 0.05 ^b^	0.40 ± 0.04 ^a^

Different letters in the same row for the same variety of quinoa indicate significant differences (*p* < 0.05).

## Data Availability

The data supporting the results of this study are included in the present article.

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
