# Peer review of "Germination of White and Red Quinoa Seeds: Improvement of Nutritional and Functional Quality of Flours"

_foods, 2022, doi:10.3390/foods11203272_

Round 1
Reviewer 1 Report
General comments:
In this study, the effect of germination on the nutritional and functional quality of quinoa were investigated. The reliable test results have certain reference value, the article content is rich and interesting, but the article still needs to correct some problems.
Specific comments:
1. What is the name of NASA, CELSS and FAD? Please use their whole name when you first used them.
2. The authors should explain the different changes of the composition of germinated quinoa when compared with the results of Padmashree et al. (11) and Antezana et al. (28). Please add some discussion about this differences.
3. Page 13: The authors just concluded from the SDS-PAGE results that the protein structure of white quinoa is different from that of red. This conclution was too hasty. The authors should carry out more research on the structual changes of pretein in the white quinoa and red quinoa.
4. Does the germination conditions affect the changes of compositon, nutritional and functional quality of quinoa during the germination? Why did the authors choose this condition for the germination of quinoa in this study? I adivise the authors could investigate the effect of different germinsation conditions on the nutritional and functional quality of quinoa. In this study, the authours should add some disscution about this aspects.
Author Response
"Please see the attachment"

Reviewer 2 Report
The manuscript presents interesting and valuable work, which is within the scope of the Foods. However, many issues should be clarified, and the structure of the paper, as well as the methods employed, is sometimes not clear. Therefore, this paper needs significant revision to be accepted for publication. I encourage the authors to consider making the necessary changes.
In my opinion, much work has been done, but the submitted manuscript should be extensively revised. The most critical part of the manuscript is the Results part, and the scientific contribution is very poorly presented, so it looks like an excellent laboratory report.
Please, the authors have to follow the instructions for the authors. All lines should use the same font.
"A calibration curve containing different concentrations (0-99 μg/mL; r2 = 0.999)" According to the ICH guidelines, the linearity requirement is a correlation coefficient (r)> 0.999, not a determination coefficient (R2). Therefore, the authors should calculate the correlation coefficient and change it.
Unfortunately, many mistakes affect the reading and following of this manuscript. A number of typing errors should be corrected.
- "as well as on quinoa in particular [10] [11] [12] [13]; however, there" ([10-13])
- „2.3.1. Proximal composition." (without full stop)
- „Eliasson [16]." (not bold)
- „2.4.3.2. Free radical scavenger activity on 2,2-diphenyl-1-picrylhydrazyl (DPPH)." (not bold)
- It is enough to write the abbreviation once: e.g., 2,2-diphenyl-1-picrylhydrazyl (DPPH)
- Units of measurement should be written uniformly according to the instructions for the author, e.g. "mL"
Uniformly write the composition of solutions
"in different buffers: (A) TRIS base 0.086 M, glycine 0.090 M and EDTA 0.003 M as extraction buffer (EB) at pH 8"
"buffer (0.5 M Tris-Base, 50% glycerol, 0.4% SDS, 5% 2-mercaptoethanol and 0.05% bromophenol blue)"
Tables are not appropriate for a manuscript (e.g. Table 1).
Figure 2. is not clear. Please, correct it.
This manuscript does not contain part of the Conclusions. Please, write conclusions.
Author Response
"Please see the attachment"

Round 2
Reviewer 1 Report
The manuscript is well-organized and well written for the publication.
Author Response
Reviewer 1
Comments and Suggestions for Authors
The manuscript is well-organized and well written for the publication.
Response: We appreciate the Reviewer comment.
Reviewer 2 Report
The authors have subsequently improved the manuscript according to most of the reviewers' requests. The scientific quality of the work described in the manuscript is generally satisfactory. Therefore, I recommend the publication of the submitted manuscript.
Author Response
Reviewer 2
Comments and Suggestions for Authors
The authors have subsequently improved the manuscript according to most of the reviewers' requests. The scientific quality of the work described in the manuscript is generally satisfactory. Therefore, I recommend the publication of the submitted manuscript.
Response: We appreciate the Reviewer comment.